# Autoreactivity against Denatured Type III Collagen Is Significantly Decreased in Serum from Patients with Cancer Compared to Healthy Controls

**DOI:** 10.3390/ijms24087067

**Published:** 2023-04-11

**Authors:** Christina Jensen, Patryk Drobinski, Jeppe Thorlacius-Ussing, Morten A. Karsdal, Anne-Christine Bay-Jensen, Nicholas Willumsen

**Affiliations:** Nordic Bioscience, 2730 Herlev, Denmark

**Keywords:** autoimmunity, collagen, extracellular matrix, cancer, biomarker, immunotherapy, immune-related adverse events

## Abstract

Autoantibodies have the potential as cancer biomarkers as they may associate with the outcome and immune-related adverse events (irAEs) following immunotherapy. Cancer and other fibroinflammatory diseases, such as rheumatoid arthritis (RA), are associated with excessive collagen turnover leading to collagen triple helix unfolding and denaturation with exposure of immunodominant epitopes. In this study, we aimed to investigate the role of autoreactivity against denatured collagen in cancer. A technically robust assay to quantify autoantibodies against denatured type III collagen products (anti-dCol3) was developed and then measured in pretreatment serum from 223 cancer patients and 33 age-matched controls. Moreover, the association between anti-dCol3 levels and type III collagen degradation (C3M) and formation (PRO-C3) was investigated. Anti-dCol3 levels were significantly lower in patients with bladder (*p* = 0.0007), breast (*p* = 0.0002), colorectal (*p* < 0.0001), head and neck (*p* = 0.0005), kidney (*p* = 0.005), liver (*p* = 0.030), lung (*p* = 0.0004), melanoma (*p* < 0.0001), ovarian (*p* < 0.0001), pancreatic (*p* < 0.0001), prostate (*p* < 0.0001), and stomach cancers (*p* < 0.0001) compared to controls. High anti-dCol3 levels were associated with type III collagen degradation (C3M, *p* = 0.0002) but not type III collagen formation (PRO-C3, *p* = 0.26). Cancer patients with different solid tumor types have downregulated levels of circulating autoantibodies against denatured type III collagen compared to controls, suggesting that autoreactivity against unhealthy type III collagen may be important for tumor control and eradication. This autoimmunity biomarker may have the potential for studying the close relationship between autoimmunity and cancer.

## 1. Introduction

Autoimmune diseases, such as rheumatoid arthritis (RA), and cancer are major health problems worldwide with challenging clinical circumstances for patients. Both diseases are driven by an imbalance and dysregulation of the immune system, but where the opposing forces may drive disease progression.

One of the main overlapping and essential processes—a common denominator of autoimmune diseases and cancer—is connective tissue remodeling resulting from fibrosis, a state of chronic inflammation, and ongoing wound healing processes [1,2]. Chronic connective tissue turnover manifests by increased synthesis and degradation of the extracellular matrix (ECM), of which collagens are major components [1,2]. Collagens are degraded by collagenolytic proteases, such as matrix metalloproteinases (MMPs), leading to triple helix unfolding and denaturation with exposure of potential immunodominant epitopes [3]. Denatured collagen is collagen where the triple helices are no longer thermally stable and unfold, resulting in denatured collagen fragments. Denatured collagen fragments can be generated by enzymes, such as MMPs, heat, mechanical tension, or pH [3,4,5]. This tissue remodeling may lead to a vicious cycle with the development of autoreactivity and autoantibodies against denatured collagen products, thereby maintaining unwanted inflammation and collagen turnover [1]. In autoimmune diseases, autoreactive lymphocytes and the initiation of the production of autoantibodies against self-epitopes and auto-antigens leads to an abnormal and unwanted immune response in an otherwise functioning body part with associated tissue destruction and, ultimately, the loss of organ function [6,7]. During cancer, the immune system is suppressed, and lymphocytes that should otherwise eliminate the tumor present with an exhausted and inactive phenotype [8,9]. This may allow a primary tumor and cancer cells to grow uncontrolled and metastasize to other organs.

Despite increased connective tissue turnover in the tumor microenvironment, few studies have investigated how immune cells, such as B cells, respond to the accumulation and degradation of collagen in cancer tissues. Ingrosso et al. have shown that autoantibodies against type I and III collagen are elevated in prostate cancer patients compared to healthy controls [10]. In line with this, serum levels of antibodies binding type I, II, III, and V collagen have been shown to be significantly higher in lung cancer patients compared to healthy controls [11].

In this study, we aimed to develop a novel biomarker assay to quantify autoantibodies against denatured type III collagen products (anti-dCol3) and investigate the balance of tissue-related autoreactivity in cancer. Interestingly, we showed that anti-dCol3 is significantly lower in serum from patients diagnosed with cancer originating from 12 different solid tumor types when compared to healthy controls. These findings are in alignment with the lack of tissue destruction (i.e., tumor eradication) found in cancer. These results may provide means for studying and monitoring the close relationship between autoimmunity and cancer, including response to cancer immunotherapy.

## 2. Results

### 2.1. Technical Evaluation of the Developed Anti-dCol3 Assay

The technical performance of the anti-Col3 assay was evaluated through different technical validation steps, and the results are summarized in Table 1. Based on 10 independent day-to-day assay runs, the inter- and intra-assay variations were determined to be 15% and 7%, respectively (acceptance criteria ≤ 15% for mean inter- and ≤10% for mean intra-assay variation [12]). The detection range was determined to be 2.8–96.5 RU/mL. Linearity was detected from initial serum dilution (1:100) to a 1:32 dilution (1:3200) with a mean dilution recovery at 103%. The analyte stability was acceptable after four freeze–thaw cycles, with a mean analyte recovery of 98%. The analyte was stable up to 48 h when stored at 4 °C or up to 4 h when stored at 20 °C. The prolonged storage of HRP-detection antibodies was acceptable up to 24 h at 20 °C. No interference was detected from low to high contents of lipids, with a mean recovery of 89% and 81%, respectively. Furthermore, no interference was observed with low and high biotin concentrations with a mean recovery of 91% and 80%, respectively. Anti-dCol3 showed interference with hemoglobin, which increased the signal, exceeding 120% mean recovery, suggesting that measured serum must be free from hemoglobin content before use.

### 2.2. Anti-dCol3 Assay Sensitivity and Specificity

First, it was investigated if thermal denaturation of type III collagen compared to native protein could increase the sensitivity of the developed anti-dCol3 assay. The testing of serum samples from patients with rheumatoid arthritis (RA) (RA1-RA5) displayed substantially higher binding of autoantibodies to the assay plates when coated with denatured rather than native type III collagen and measured with the anti-dCol3 assay (mean increase, 4.7-fold; range, 2.0–10.3-fold) (Table 2). The anti-dCol3 assay was coated with albumin with a molecular ratio equal to collagen as a non-sense control, and without protein as a background control. The signal was 7.8-fold and 7.0-fold higher when plates were coated with denatured collagen compared to albumin or without protein, respectively (Table 2). Based on these results, the anti-dCol3 assay was coated with denatured type III collagen.

The specificity and sensitivity of the developed anti-dCol3 assay were investigated further by overnight incubation of five serum samples from RA patients (RA1-RA5) with 10 ug/mL of denatured collagen, native collagen, and albumin (Table 2). The hypothesis behind this experiment was that the spiking with denatured collagen will compete for binding to the autoantibodies in the RA serum, resulting in a decreased signal. Incubation with native collagen was introduced as a control, and albumin was included as a non-sense control. Compared to the unspiked serum samples, a mean signal decrease of 67.0% (range, 56.7–78.0%) was observed if the serum was spiked with denatured type III collagen (Table 2). When incubating the samples with native type III collagen, a mean signal decrease of 22.0% (range, 11.5–29.0%) was observed compared to the unspiked serum samples (Table 2). These data suggest more extensive anti-collagen III autoantibodies binding to the denatured compared to the native protein. Furthermore, serum spiking with albumin did not result in a difference compared to the unspiked serum (mean signal decrease of −6.6%) (Table 2), suggesting anti-dCol3 assay specificity for the detection of anti-collagen III autoantibodies.

### 2.3. Biological Evaluation of the Anti-dCol3 Assay in Cancer and RA

Next, the anti-dCol3 levels were evaluated in RA (n = 5) and cancer samples (n = 223) compared to healthy controls (n = 33). Anti-dCol3 levels were 1.9-fold higher in serum from RA patients compared to healthy controls (Figure 1a). Statistical analysis was not performed as the number of RA patients was too small. Interestingly, when assessing anti-dCol3 in cancer patients, anti-dCol3 levels in serum were significantly decreased in bladder cancer (*p* = 0.0007), breast cancer (*p* = 0.0002), colorectal cancer (*p* < 0.0001), head and neck cancer (*p* = 0.0005), kidney cancer (*p* = 0.005), liver cancer (*p* = 0.030), lung cancer (*p* = 0.0004), melanoma (*p* < 0.0001), ovarian cancer (*p* < 0.0001), pancreatic cancer (*p* < 0.0001), prostate cancer (*p* < 0.0001), and stomach cancer (*p* < 0.0001) compared to healthy controls (Figure 1b), on average, resulting in a 62% reduction in levels of circulating anti-dCol3 across all solid tumor types compared to healthy controls. When evaluating anti-dCol3 in cancer patients with different cancer stages, no association with stages was observed (Appendix A).

To investigate the diagnostic accuracy of anti-dCol3 for discriminating between all cancer patients and healthy controls, ROC analysis was performed. The ROC analysis was conducted using 223 cancer patients vs. 33 healthy controls. The anti-dCol3 assay provided a high diagnostic power for discriminating between cancer and healthy controls, with an AUROC of 0.88, *p* < 0.0001 (Figure 1c).

### 2.4. Evaluation of Circulating Type III Collagen Fragments (C3M and PRO-C3) and Their Association with the Anti-dCol3 Assay

Next, we evaluated other type III collagen biomarkers in the cancer patients compared to healthy controls and investigated their association with the type III collagen autoantibodies (anti-dCol3). MMP-mediated type III collagen degradation (C3M) was significantly elevated in cancer patients compared to healthy controls (*p* < 0.0001), whereas type III collagen formation (PRO-C3) was borderline elevated in the cancer patients (*p* = 0.063) (Figure 2a,b).

Then, we evaluated the association between anti-dCol3 and C3M and PRO-C3 by dividing patients into quartiles based on their C3M or PRO-C3 levels, respectively. Interestingly, higher anti-dCol3 levels were significantly associated with higher C3M levels in a stepwise manner: Q1 < Q2 < Q3 < Q4 (*p* = 0.0002), whereas no association was observed between PRO-C3 and anti-dCol3 levels: Q1 = Q2 = Q3 = Q4 (*p* = 0.26) (Figure 2c,d).

## 3. Discussion

In the present study, we developed an assay to detect autoantibodies against denatured type III collagen (anti-dCol3) and evaluated its biomarker potential in cancer. The main findings were that (1) a technically robust assay specific for autoantibodies against denatured type III collagen (anti-dCol3) was developed, (2) anti-dCol3 was significantly decreased in 12 types of cancer compared to healthy controls, (3) anti-dCol3 was associated with MMP-mediated type III collagen degradation (C3M). To our knowledge, this is the first study to show reduced autoreactivity against denatured type III collagen in cancer.

Interestingly, the anti-dCol3 assay was shown to only detect autoantibodies against denatured type III collagen and not native collagen. This is further supported by the data showing that anti-dCol3 levels are associated with MMP-9 cleavage products of denatured type III collagen (C3M) but not type III collagen formation (PRO-C3), suggesting an association with MMP-mediated tissue remodeling [3,5,13]. As expected, based on previous studies of RA patients showing autoantibodies directed against type III collagen [14,15], anti-dCol3 was almost two-fold higher in serum from patients with RA compared to healthy controls (Figure 1a), which could reflect increased autoreactivity and autoantibodies against these collagen products leading to unwanted inflammation and tissue destruction. Conversely, the lower anti-dCol3 levels in the different solid tumor types compared to healthy controls (Figure 1b) could reflect a decreased IgG-mediated clearance of type III collagen fragments and a lack of tissue destruction/tumor eradication by the immune system. Cancer is a fibrotic disease characterized by excessive collagen deposition in the tumor microenvironment, which is associated with tumor progression and poor patient prognosis [16,17,18]. This study raises an interesting question of whether increased cancer fibrosis is actually a consequence of decreased collagen autoimmunity and clearance of collagen fragments. This link should be further investigated in future studies, as it could provide important information for cancer drug development.

Opposite to our results, previous studies have observed higher levels of autoantibodies against type III collagen in prostate and lung cancer patients compared to healthy controls [10,11]. In these studies, the autoantibodies detect native type III collagen, whereas anti-dCol3 in our study measures denatured collagen, which may explain the novel finding of low autoantibodies in cancer patients in our study. The level of anti-dCol3 found in healthy controls may represent a balanced level of autoantibodies and clean-up of unwanted denatured type III collagen degradation fragments [19]. If this balance is shifted in one direction (increased), it is associated with diseases such as RA, whereas if the balance is shifted in the other direction (decreased), it is associated with diseases such as cancer.

In the cancer immunotherapy setting, the balance in the activation of the immune system is also a critical step where tumor cells, which normally are recognized by T cells, have developed strategies to take advantage of peripheral tolerance and immune checkpoints [20]. The development of immune checkpoint inhibitors such as anti-PD-1 has revolutionized the treatment of several solid tumor types such as melanoma and lung cancer because of the possibility of long-term responses [21,22]. However, despite the success of immune checkpoint inhibitors, it has proven to be difficult to reactivate T cell immunity without activating unwanted autoimmunity, and a majority of patients experience immune-related adverse events (irAEs) [23]. Even though this has been a huge problem in the last decade, irAEs are still not well understood, and biomarkers for the prediction of the development of irAEs are lacking. IrAEs are caused by the nonspecific activation of the immune system and can occur in any organ system, where the most common irAEs are hypothyroidism, thyrotoxicosis, autoimmune hepatitis, colitis, neuropathy, arthralgia, and psoriasis [24]. As we observed the opposite with higher anti-dCol3 levels in the autoimmune RA disease patients and lower anti-dCol3 levels in cancer patients in this study, this novel anti-dCol3 autoimmunity biomarker may have the potential for studying and monitoring the close relationship between autoimmunity and cancer, such as the risk of developing irAEs on immune checkpoint inhibitors or developing cancer on rheumatoid arthritis immunosuppressants.

The present study is limited in that the anti-dCol3 biomarker was only measured in 20 patients with different cancer types and only three liver cancer patients. Moreover, it is a limitation that anti-dCol3 was only measured in five RA patients, making it difficult to apply statistics. Furthermore, the exact epitope or epitopes on the denatured type III collagen recognized by the anti-dCol3 autoantibody are not yet discovered, which could provide unique value and should be investigated further. This study was a proof-of-concept study, which needs to be validated using larger patient cohorts. Moreover, it could be interesting to investigate the anti-dCol3 biomarker in an immune checkpoint inhibitor study where we also have information about irAEs.

## 4. Materials and Methods

### 4.1. Development and Procedure of the Anti-dCol3 Assay

The development of the anti-dCol3 assay was preceded by experiments that aimed to find the most optimal reagents, protein concentrations, reaction times, and temperatures [13,25]. The developed anti-dCol3 assay utilizes a commercial mouse monoclonal anti-human HRP-labelled antibody of the IgG1 isotype (Abcam, Berlin, Germany, ab99759). The antibody is used for the specific detection of IgG autoantibodies targeting type III collagen in human serum. According to the manufacturer, Abcam antibodies specifically react with the Fc portion of the heavy chain of all subclasses of human IgG (IgG1–IgG4), as demonstrated by ELISA [26]. In addition, the antibodies demonstrated no levels of cross-reactivity with other antibody isotypes. The antibodies were purified using protein G chromatography and displayed a high level of purity. The optimized ELISA protocol for the anti-dCol3 assay is comprised of the following steps: native human type III collagen (Abcam, ab7535) was initially pre-heated in an Eppendorf ThermoMixer (Hørsholm, Denmark) for 15 min at 72 °C. Pre-heated denatured collagen was subsequently diluted in carbonate coating buffer (pH = 9.5) to a final concentration of 1 µg/mL. A 96-well white microplate (Nunc MaxiSorp™, Fisher Scientific, Roskilde, Denmark) was coated with 100 µL/well of prepared denatured type III collagen for 20 h at 4 °C with shaking at 300 rounds per minute (rpm). The plate was subsequently washed five times with washing buffer (25 mM Tris, 50 mM NaCl, 0.1% Tween20, pH = 7.2) using an automated microplate washer and blocked with 100 µL/well of Chonblock blocking buffer (Chondrex, Woodinville, WA, USA) for 2 h at 20 °C with shaking at 300 rpm. After the completed incubation, the plate was washed five times with a washing buffer. Serum samples were diluted (1:100) in assay buffer (50 mM PBS-BTB, 8 g NaCl, pH = 7.4), and 100 µL of each were applied in duplicates, together with standard and controls, and incubated for 2 h at 20 °C with shaking at 300 rpm. The plate was again washed five times with washing buffer. Then, 100 µL/well of mouse monoclonal anti-human IgG HRP-antibodies (Abcam, ab99759) at a concentration of 200 ng/mL diluted in assay buffer were applied and incubated for 1 h at 20 °C with shaking at 300 rpm. After the completed incubation, the plate was washed five times with a washing buffer. Subsequently, 100 µL/well of prepared BM Chemiluminescence ELISA POD substrate (Roche, Mannheim, Germany) was added, and the plate was immediately placed in a Fluoroskan Microplate Reader (Thermofisher, Roskilde, Denmark). The plate was evaluated with Thermo Scientific SkanIt software version 6.0.1 (Thermofisher) in luminescence mode and with the following parameters: 1 min of shaking at 240 rpm (medium speed), 2 min of pause, and a measurement time of 1000 ms. A standard curve was plotted using a 5-parameter logistic curve fit (Appendix A).

Pre-heated type III collagen was used to coat the plates, as it was observed that thermal denaturation of type III collagen compared to native protein increases the sensitivity of the developed anti-dCol3 assay. This was investigated by coating the plates for 20 h at 4 °C with shaking with 1 ug/mL pre-heated type III collagen (denatured collagen) (Abcam, ab7535), 1 ug/mL native type III collagen (Abcam, ab7535), or 1 ug/mL albumin (Sigma-Aldrich, A6608). Afterward, the ELISA protocol was followed as above.

### 4.2. Technical Evaluation of the Anti-dCol3 Assay

The reproducibility of the assay was established by determining the inter- and intra-assay coefficients of variation (CV). Inter-assay variation was acceptable if CV was ≤15%, and intra-assay variation was acceptable if CV ≤ 10% [12]. Assay variation was established based on 10 independent day-to-day runs, including standard, two assay controls (one below and one above assay EC50), and 5 lot-to-lot samples covering the entire analytical measurement range. Quantification of antibodies against denatured type III collagen in human serum was based on a 2-fold diluted 7-point standard curve. Both the standard curve and controls were developed from the native serum material from RA patients with high disease activity. The measurement range of the assay was defined as the lower limit of measurement range (LLMR) and the upper limit of measurement range (ULMR) and calculated from 10 day-to-day independent runs. As no international reference units exist for the quantification of antibodies against collagen in serum, the calibration was performed in relative units (RU/mL). Each analytical run was accepted only if all standard points displayed CV ≤ 10% and the percentage of relative error (%RE) ≤15% within the analytical measurement range. Similarly, all assay controls needed to display CV ≤ 20% with a calculated concentration within the target range of mean ± 20%. All tested samples were run in duplicates and accepted if CV was ≤20%. The linearity of the assay was determined by a dilution recovery test based on serial dilutions (2-fold) of four serum samples (from 1:100 to 1:3200). The analyte stability was assessed by temperature treatment and four freeze and thaw cycles of three serum samples. Temperature treatment involved serum incubation at 0, 2, 4, 24, and 48 h at either 4 °C or 20 °C. Moreover, analytical interference with common serum components was assessed by the addition of low/high concentrations of hemoglobin (2.5/5 mg/mL), lipids (1.5/5 mg/mL), or biotin (3.0/9.0 ng/mL) to serum samples. Untreated serum was used as a reference point for the calculation of the recovery percentage. Lastly, denatured or native type III collagen was spiked into serum samples to investigate its impact on the anti-dCol3 measurements.

### 4.3. Biological Evaluation of the Anti-dCol3 Assay in Cancer Patients Compared to Healthy Controls

The biological relevance of the anti-dCol3 assay was assessed in serum from patients with RA and with different solid tumor types and compared to healthy controls (Table 3).

A small cohort of five RA patients was included in this study and consisted of patients from the AMBITION study, a phase III trial (NCT00109408), where they received tocilizumab monotherapy (8 mg/kg intravenously every 4 weeks). Anti-dCol3 was assessed in serum from baseline from RA patients that had moderate-to-severe RA for ≥3 months. Patient inclusion and exclusion criteria were described previously [27]. RA patient demographics are shown in Table 3.

A cohort of 223 cancer subjects was included in this study and consisted of patients with bladder cancer (n = 20), breast cancer (n = 20), colorectal cancer (n = 20), head and neck cancer (n = 20), kidney cancer (n = 20), liver cancer (n = 3), lung cancer (n = 20), melanoma (n = 20), ovarian cancer (n = 20), pancreatic cancer (n = 20), prostate cancer (n = 20), and stomach cancer (n = 20). All cancer samples were obtained from Proteogenex (Los Angeles, CA, USA). Additionally, a control cohort involved age-matched healthy controls (n = 33) obtained from BioIVT (Westbury, NY, USA) was included. Serum sample collection was approved by an Institutional Review Board or Independent Ethical Committee, and patients gave their informed consent (protocol number: PG-ONC 2003/1). All investigations were carried out according to the Helsinki Declaration. Serum samples were collected pretreatment. Cohort demographics are shown in Table 4.

To investigate the association of anti-dCol3 with other type III collagen biomarkers, type III collagen degradation (C3M) and type III collagen formation (PRO-C3) were also measured in cancer patients. C3M and PRO-C3 are well characterized, competitive ELISAs based on a monoclonal antibody specific toward an MMP-9-generated neo-epitope on type III collagen (C3M) and a monoclonal antibody specific for the N-proteinase cleavage site of the N-terminal pro-peptide of type III collagen (PRO-C3) [13,25]. The ELISAs are manufactured by Nordic Bioscience (Herlev, Denmark), and measurements were performed according to the manufacturer’s specifications [13,25].

### 4.4. Statistical Analysis

The difference between healthy controls and RA or the combined cancer group was tested using the Mann–Whitney test. The difference between healthy controls and the different cancer groups was tested using the nonparametric Kruskal–Wallis test with Dunn’s multiple comparisons tests. AUROC analysis was performed to evaluate the diagnostic power for discriminating between cancer patients and healthy controls. A *p*-value of <0.05 was considered statistically significant. All statistical analyses and graphical presentations were performed using GraphPad version 9.5.1 (GraphPad Software, Inc., San Diego, CA, USA).

## 5. Conclusions

Autoimmune diseases and cancer are characterized by increased remodeling and degradation of collagen. While autoimmunity against denatured collagen fragments is a well-known consequence of this tissue remodeling in autoimmune diseases, resulting in unwanted inflammation and tissue destruction, not much is known about autoimmunity against collagen in cancer patients. We developed a technically robust serological biomarker that measures autoantibodies against denatured type III collagen fragments (anti-dCol3) and investigated its biomarker potential in cancer. This is the first study to show reduced autoreactivity against denatured type III collagen in cancer, which could reflect a decreased clean-up of self type III collagen degradation neo-epitopes and a lack of tissue destruction/tumor eradication by the immune system, leading to tumor fibrosis and tumor progression. This explorative study suggests that this non-invasive autoimmunity biomarker (anti-dCol3) has the potential for studying and monitoring the close relationship between autoimmunity and cancer.

## 6. Patents

A patent has been filed for the anti-dCol3 assay.

## Figures and Tables

**Figure 1 ijms-24-07067-f001:**
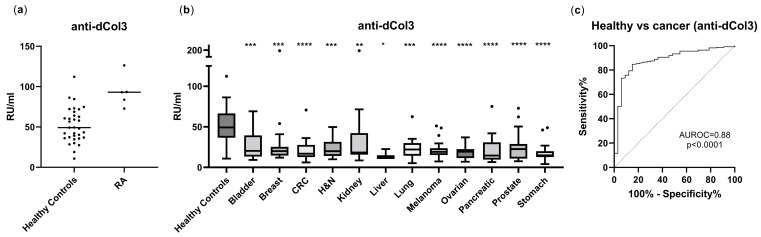
Anti-dCol3 levels were significantly decreased in 12 types of cancer compared to healthy controls. (**a**) Anti-dCol3 levels in rheumatoid arthritis (RA) patients (n = 5) compared to healthy controls (n = 33). The line represents the median. (**b**) Anti-dCol3 levels in patients with bladder cancer (n = 20), breast cancer (n = 20), colorectal cancer (n = 20), head and neck cancer (n = 20), kidney cancer (n = 20), liver cancer (n = 3), lung cancer (n = 20), melanoma (n = 20), ovarian cancer (n = 20), pancreatic cancer (n = 20), prostate cancer (n = 20), and stomach cancer (n = 20) compared to healthy controls (n = 33). Biomarker levels are presented as Tukey box plots, and statistical differences were analyzed using the Kruskal–Wallis test adjusted for Dunn’s multiple comparisons tests. * *p* < 0.05, ** *p* < 0.01, *** *p* < 0.001, **** *p* < 0.0001. (**c**) ROC analysis for identification of diagnostic power of anti-dCol3 for discrimination between cancer patients and healthy controls.

**Figure 2 ijms-24-07067-f002:**
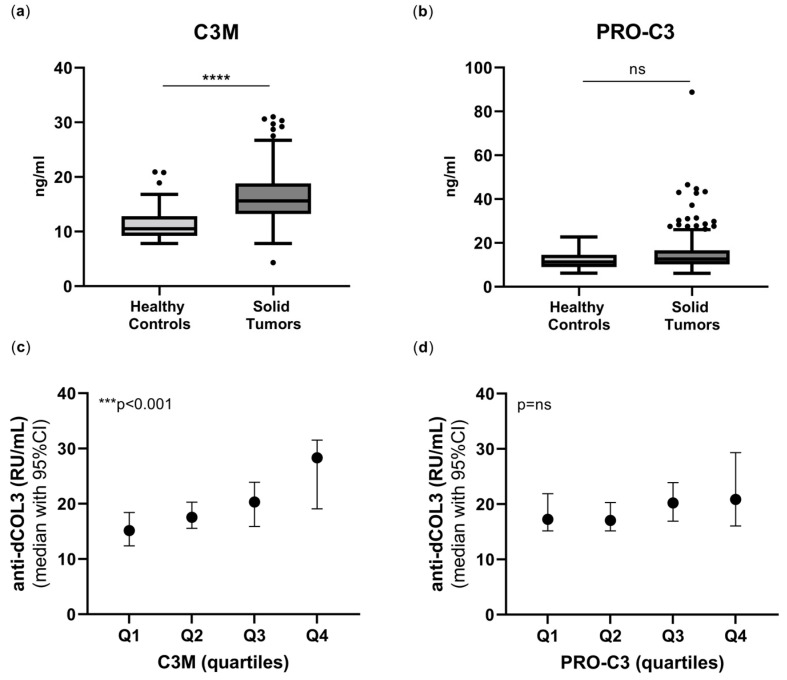
Type III collagen degradation (C3M) is elevated in cancer and associated with anti-dCol3. (**a**) MMP-mediated degradation of type III collagen (C3M) and (**b**) formation of type III collagen (PRO-C3) in 223 cancer patients compared to healthy controls (n = 33). Biomarker levels are presented as Tukey box plots, and statistical differences were analyzed using the Mann–Whitney test. (**c**) C3M and (**d**) PRO-C3 levels in the cancer patients were divided into quartiles, and the anti-dCol3 levels presented as median with 95% CI were compared between each quartile with a Kruskal–Wallis test adjusted for Dunn’s multiple comparisons tests. ns: non-significant, *** *p* < 0.001, **** *p* < 0.0001.

**Table 1 ijms-24-07067-t001:** Technical performance of the anti-dCol3 assay.

Assay Parameters	Results
Inter-assay variation, mean (range)	15% (9–28%)
Intra-assay variation, mean (range)	7% (4–9%)
Measurement range (RU/mL) LLMR-ULMR	2.8–96.5
EC50, mean (range)	40.8 (33.9–46.9)
Std A concentration (RU/mL)	100
Slope, mean (range)	1.03 (0.81–1.44)
Dilution recovery of human serum, mean (range)	103% (74–119%)
Analyte stability	Up to 48 h at 4 °C or 4 h at 20 °C
Analyte recovery, 4 freeze–thaw cycles, mean (range)	98% (89–104%)
Antibody stress	Up to 24 h at 20 °C
Hemoglobin recovery	>120%
Lipids Recovery (low)(high)	(89%)(81%)
Biotin Recovery (low)(high)	(91%)(80%)

LLMR: lower limit of measurement range, ULMR: upper limit of measurement range.

**Table 2 ijms-24-07067-t002:** Anti-dCol3 assay sensitivity and specificity.

	RA1	RA2	RA3	RA4	RA5	Mean
**Fold change when coating with denatured collagen compared to**						
Native collagen	3.9	10.3	2.7	4.7	2.0	4.7
Albumin	4.8	15.2	4.8	11.2	2.8	7.8
Background	6.3	13.5	3.0	9.1	3.0	7.0
**Signal decrease in % compared to unspiked serum**						
Denatured collagen	61.9	78.0	64.9	73.3	56.7	67.0
Native collagen	29.0	27.7	23.6	18.5	11.5	22.0
Albumin	−5.4	14.1	2.8	−0.7	−43.6	−6.6

RA: rheumatoid arthritis.

**Table 3 ijms-24-07067-t003:** RA cohort demographic.

**RA Disease (Mean, (SD))**	
Disease activity score	7.3 (1.1)
Tender joint count	46.4 (19.7)
Swollen joint count	38.0 (23.0)
**Age (years)**	
Mean (SD)	57.6 (12.0)
Median (min, max)	58 (42, 75)
Missing	0 (0%)
**Sex**	
Male	0 (0%)
Female	5 (100%)
**Ethnicity**	
Non-hispanic	3 (60%)
Hispanic	2 (40%)

RA: rheumatoid arthritis.

**Table 4 ijms-24-07067-t004:** Cohort demographics of cancer patients and healthy controls.

	Healthy (N = 33)	Cancer (N = 223)	Total (N = 256)
**Diagnosis**			
Healthy	33 (100%)		33 (12.9%)
Bladder cancer		20 (9.0%)	20 (7.8%)
Breast cancer		20 (9.0%)	20 (7.8%)
CRC		20 (9.0%)	20 (7.8%)
H&N cancer		20 (9.0%)	20 (7.8%)
Kidney cancer		20 (9.0%)	20 (7.8%)
Liver cancer		3 (1.3%)	3 (1.2%)
Lung cancer		20 (9.0%)	20 (7.8%)
Melanoma		20 (9.0%)	20 (7.8%)
Ovarian cancer		20 (9.0%)	20 (7.8%)
Pancreatic cancer		20 (9.0%)	20 (7.8%)
Prostate cancer		20 (9.0%)	20 (7.8%)
Stomach cancer		20 (9.0%)	20 (7.8%)
**Stages**			
I		7 (3.1%)	7 (2.7%)
II		49 (22.0%)	49 (19.1%)
III		93 (41.7%)	93 (36.3%)
IV		74 (33.2%)	74 (28.9%)
**Age (years)**			
Mean (SD)	57.7 (5.69)	59.3 (11.2)	59.1 (10.7)
Median (min, max)	57.0 (49.0, 69.0)	61.0 (30.0, 87.0)	60.0 (30.0, 87.0)
Missing	0 (0%)	1 (0.4%)	1 (0.4%)
**Sex**			
Male	21 (63.6%)	121 (54.3%)	142 (55.5%)
Female	12 (36.4%)	102 (45.7%)	114 (44.5%)
**Ethnicity**			
Black	13 (39.4%)	0 (0%)	13 (5.1%)
Caucasian	11 (33.3%)	223 (100%)	234 (91.4%)
Hispanic	9 (27.3%)	0 (0%)	9 (3.5%)

CRC: colorectal cancer, H&N: head and neck cancer.

## Data Availability

The data presented in this study are available on request from the corresponding author.

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
