# Peer review of "Autoreactivity against Denatured Type III Collagen Is Significantly Decreased in Serum from Patients with Cancer Compared to Healthy Controls"

_ijms, 2023, doi:10.3390/ijms24087067_

Round 1
Reviewer 1 Report
Jensen et al. demonstrated the detection of autoantibody on denatured type 3 collagen and its relationship with cancer and rheumatoid arthritis patients. Please refer to the comments below:
1) Title: The title does not reflect the work done and it sounds complicated. Please amend it and make it sounds straightforward. Please include RA in the title as well
2) Abstract: Some comments for the methodology & results. Update the abstract accordingly
3) Result: Please cite to support those acceptance criteria including the CV & recovery)
4) Result: Kindly explain as the authors mentioned the accepted inter-assay CV should be LESS than 15%, but you got it at 15% [9-28%].
5) Line 121-124, 196 & Table 2: This work (native collagen & albumin coating) was not described in the methodology. The authors only described using denatured type 3 collagen in coating (line 288-293).
6) Line 129: Is this (2x1013/ml) a typing error?
7) Line 167-168: If statistical analysis being done, please include it as supplementary data.
8) Figure 1C: Please describe how the authors performed ROC analysis, like from which software.
9) Table 2: Please include the basic demographic and RA stage
10) Line 280-285: Please cite the manufacturer (website or instruction manual)
11) Line 287-312: Please cite if the authors are referring to some previous work. Please mention if this is from in-house optimization steps.
12) Line 311: Please include the standard curve as a supplementary data.
13) Line 300: As sample dilution, a 100x was used. How this is different from the dilution recovery test (line332)? Dilution recovery test used undiluted serum in starting?
14) Discussion: Please include the limitations of having only 5 RA patients and 3 liver cancer patients. This definitely will affect the statistical power due to the extremely small sample size.
15) Table 3: Kindly update the table 3 where % for different type of cancer is not necessary; please leave it blank for "cancer type" and "cancer stage" under Healthy subjects.
16) Please update the abstracts, Introduction, discussion, and conclusion as the current finding does not directly answer the risk of developing irAEs
Reviewer 2 Report
In the current manuscript, the authors need to introduce what is denatured collagen. How it’s possible to be denatured collagen is in vivo? Is denatured collagen very different then MMP chewed collagens, or do these denatured collagens have something to do with whole collagen with altered biophysical properties? What immune response is associated with denatured collagens vs. fragmented collagen; Th dependent, Tc-dependent, B cell-dependent or macrophage dependent?
Information about serum needs to be put by authors. Cohorts of RA studies, affiliated protocol number. Size of populations (race, age, ethnicity etc.). part of any clinical trial etc., with information about RA stage vs. treatments bDMARDS and DMARDS exposure. Any info about ADA (Antibody-Dependent antibody response). Any info about Rituximab treated samples and anti COL3 antibody?
Please provide the rationale or proof that anti-dCol3 will recognize enzymatic digested collagen 3 (COL3) (tryptic or chymotryptic digested collagen fragment) with the same efficiency as denatured COL3.
Authors do not want to stick with one condition; authors are switching between cancer and RA without the context. Please decide your belief system, and please articulate which disease you want to address in this manuscript.
RA samples don’t have matching numbers (33 vs 5) I am not sure statistics can be applied here. Please justify.
Round 2
Reviewer 2 Report
Thank you for the response